# GLOBAL-AWARE EDGE PRIORITIZATION FOR POSE GRAPH CONSTRUCTION IN SFM

## ABSTRACT

The pose graph is an essential component of Structure-from-Motion (SfM) pipelines, where images form the nodes and edges encode relative poses between them. These graphs are typically sparse to reduce the cost of geometric verification required for each candidate edge. In this paper, we focus on robust pose graph initialization, performed at the very beginning of the SfM pipeline. Traditionally, this step relies on image retrieval methods applied independently to each image, connecting it to the $k$ most similar ones according to, e.g., embedding similarity. While effective in practice, this greedy approach does not allow communication across image pairs during graph construction. We address this limitation through the novel concept of edge prioritization, which ranks edges by their utility for SfM. We achieve this through the following two contributions. First, we propose an image representation network combined with a graph neural network (GNN), trained with SfM-derived supervision to predict edge ranks. The GNN exploits global context from the entire image set to guide pair selection. Second, we introduce an edge selection strategy based on minimum spanning trees, which uses predicted ranks to identify the most promising pairs. By integrating global information at both stages, our approach substantially improves SfM reconstruction in the high-speed regime, particularly when operating with very sparse pose graphs. Code will be released.

## 1 INTRODUCTION

Large-scale 3D reconstruction from unstructured image collections is a central problem in computer vision. Typically, a 3D point cloud and the camera poses are obtained from images using Structure-from-Motion (SfM) techniques (Ullman, 1979; Schonberger & Frahm, 2016). These methods estimate camera poses and 3D structure from a collection of images, enabling applications in virtual reality, visual localization (Panek et al., 2023), autonomous driving (Song & Chandraker, 2014), and novel view synthesis (Song et al., 2020; Riegler & Koltun, 2021; Kerbl et al., 2023).

SfM methods are broadly divided into two directions: incremental pipelines (Schonberger & Frahm, 2016), which iteratively expand a reconstruction by adding images and refining structure, and global pipelines (Pan et al., 2024), which estimate camera poses jointly before bundle adjustment. Despite their algorithmic differences, both approaches begin with the same core steps: construction of an initial pose graph (Barath et al., 2021), keypoint detection (Lowe, 2004; DeTone et al., 2018), feature matching (Sarlin et al., 2020; Lindenberger et al., 2023), and geometric verification (Fischler & Bolles, 1981; Barath et al., 2019). The pose graph, where cameras are nodes and edges encode relative poses, forms the structural backbone of SfM. In principle, the graph could be fully connected by verifying all $\binom{n}{2}$ image pairs, but this is computationally infeasible. Hence, constructing a sparse yet reliable pose graph is a critical step for both incremental and global pipelines.

Traditionally, pose graph initialization is based on image retrieval methods, which connect each image to its $k$ most similar neighbors based on visual descriptors (Arandjelovic et al., 2016; Berton & Masone, 2025). Although effective in practice, this approach treats each image independently and prevents information exchange across pairs during graph construction. More critically, once the initial graph is built, new edges are rarely added, but only pruned Wilson & Snavely (2014); Manam & Govindu (2024); Damblon et al. (2025). This means that important global cues, overlooked during initialization, are irretrievably lost, limiting the quality of later refinement (Zach et al., 2010; Wilson & Snavely, 2014) and ultimately restricting the accuracy of reconstruction. Thus, the quality

Figure 1: Two examples of non-matchable pairs. Each row shows (left) a query image, (middle) a hard negative like doppelganger (*top row*) or extreme viewpoint difference (*bottom*), and (right) the correct match. Numbers denote the predicted rank (lower is better). MegaLoc (Berton & Masone, 2025) incorrectly favors the doppelganger, while our method assigns the correct image a better rank.

of the initial pose graph is crucial: a stronger starting point leads to better and more efficient Structure-from-Motion downstream.

In this work, we propose to move beyond per-image retrieval and, instead, prioritize edges according to their global utility for SfM. We introduce the concept of edge prioritization, which directly ranks candidate edges by their expected contribution to reconstruction. Our approach consists of two components. First, we design a Graph Neural Network (GNN), trained with SfM-derived supervision, to predict edge ranks by globally reasoning over the entire image set. Second, we propose a multi-spanning-tree-based edge selection mechanism that uses these predicted scores to construct sparse yet well-connected pose graphs, ensuring both efficiency and robustness.

By incorporating global reasoning into both rank prediction and edge selection, our method produces more reliable pose graphs from the outset, substantially improving SfM performance in the high-speed regime, particularly when operating with very sparse graphs. As shown in Fig. 1, our approach aims at ranking reliable image pairs higher while lowering the rank of unmatchable pairs, such as doppelgangers (*top row*) and images with extreme viewpoint difference (*bottom*). Note that the extreme case is not a negative for retrieval task, but almost impossible for SfM since matching fails.

## 2 RELATED WORK

**Structure-from-Motion.** SfM reconstructs 3D structure and camera poses from image collections (Schonberger & Frahm, 2016). Incremental pipelines expand the model sequentially, registering one image at a time with repeated bundle adjustment. They are robust but can drift and scale poorly to large datasets. Global pipelines (Carlone et al., 2015; Zhu et al., 2018; Cui & Tan, 2015; Pan et al., 2024) instead estimate all poses jointly, often via motion averaging on an initial pose graph, followed by bundle adjustment. They are more efficient, and recent work such as GLOMAP (Pan et al., 2024) shows that global SfM can also reach state-of-the-art accuracy.

Regardless of strategy, both incremental and global pipelines begin with the same steps: pose graph initialization via image retrieval, local feature detection (Lowe, 2004; DeTone et al., 2018), feature matching (Sarlin et al., 2020; Lindenberger et al., 2023), and geometric verification (Fischler & Bolles, 1981; Barath et al., 2019). The initial pose graph provides the backbone: initialization proposes edges, verification assigns relative poses and prunes weak ones. Its quality is critical, since missing edges are rarely recovered later, and early errors propagate into optimization.

**Pairwise Image Similarity Learning.** Most pipelines initialize the pose graph with image retrieval, where global descriptors map visually similar images to nearby embeddings. VLAD (Jégou et al., 2010) and NetVLAD (Arandjelovic et al., 2016) are classical choices, while CosPlace (Berton et al., 2022) reframes retrieval as classification. Recent methods leverage vision foundation models: DINOv2-SALAD (Izquierdo & Civera, 2024) aggregates patch tokens via optimal transport, and MegaLoc (Berton & Masone, 2025) fine-tunes DINOv2-SALAD to achieve state-of-the-art results.

Re-ranking further improves candidate selection. Patch-NetVLAD (Hausler et al., 2021) adds region- and patch-level detail, and VOP (Wei et al., 2025) predicts overlap scores for voting-based selection.

However, these approaches operate strictly pairwise: each image pair is scored independently, ignoring global consistency. Since edges are rarely added after initialization, valuable connections can be lost permanently. This motivates global reasoning during pose graph construction.

**Pose Graph Construction and Filtering.** Beyond pairwise retrieval, several works aim to improve how pose graphs are constructed and refined. Barath et al. (2021) accelerate graph building by reusing poses from partially reconstructed graphs, while Barath et al. (2022) propose a Bayesian scheme to terminate RANSAC early on unmatchable pairs, reducing verification cost. These approaches improve efficiency at the verification stage but do not reconsider which candidate edges should be selected in the first place.

In common practice, pose graphs are built by connecting each image to its $k$ nearest neighbors based on engineered or learned embeddings (Schonberger & Frahm, 2016). While effective, this strategy often results only in local connectivity. Minimal spanning trees (MSTs) have been explored as an alternative, offering globally connected yet sparse graph priors. Roberts et al. (2011) propose sampling MSTs as minimal hypotheses to improve robustness against duplicates. Barath et al. (2021) employ a single MST for initialization and progressively add edges for stability, while Light3R-SfM (Elflein et al., 2025) also relies on a single MST and reports reduced accuracy compared to denser initialization. These works highlight a trade-off: MSTs guarantee global connectivity with minimal edges, but relying on a single tree makes the graph fragile to edge failures. Moreover, multiple spanning trees have been used in refinement stages. For instance, Xiao et al. (2021) refine pose graphs on the union of MSTs via loop consistency, and Gan et al. (2024) accelerate verification using orthogonal trees constructed from similarity matrices of engineered features. In contrast, our approach constructs *multiple* MSTs directly during initialization, guided by learned edge ranks, thereby combining complementary connectivity while maintaining sparsity.

Once the pose graph is built, filtering techniques are often applied to remove unreliable edges before motion averaging or bundle adjustment. Chen et al. (2020) cluster graphs and solve SfM in parallel subgraphs to increase efficiency. Manam & Govindu (2023; 2024) detect unstable or redundant structures by analyzing camera triplets and relative translation sensitivities. To handle ambiguities from repetitive or symmetric scenes, Doppelganger detection (Cai et al., 2023; Xiangli et al., 2025) trains classifiers to filter pairs that are visually similar but geometrically inconsistent. While effective, these methods operate only after tentative matches and geometric verification, which are computationally expensive steps our approach seeks to reduce.

Graph Neural Networks (GNNs) have recently been introduced for pose graph optimization. PoGO-Net (Li & Ling, 2021) employs message passing to denoise relative poses in an existing graph, improving rotation averaging. Brynte et al. (2024) use attention-based GNNs for 3D point tracking and pose estimation, and Damblon et al. (2025) propose a GNN to filter outlier edges. However, these methods operate on graphs where relative poses have already been estimated. By contrast, we predict the reliability of candidate edges *before* geometric verification, allowing cleaner graphs to be constructed from the outset. Combined with multiple MSTs, this yields pose graphs that are both sparse and robust, improving downstream accuracy and efficiency.

## 3 GLOBAL EDGE PRIORITIZATION

We now formalize the problem of pose graph initialization as an *edge ranking* task and present our two contributions: (i) a GNN-based model trained with geometric supervision to predict edge ranks, and (ii) a multi-spanning-tree-based edge selection strategy that outperforms traditional $k$NN heuristics. An overview of the pipeline during training and inference is shown in Fig. 2.

### 3.1 PROBLEM STATEMENT

Given $N$ input images $I = \{I_1, \dots, I_N\}$, the goal is to construct an *initial pose graph* $\mathcal{G}_0 = (\mathcal{V}, \mathcal{E}_0)$, where $\mathcal{V}$ indexes images and each candidate edge $(i, j) \in \mathcal{E}_0$ denotes an image pair that potentially has overlapping views. Since verifying all $\binom{N}{2}$ pairs is infeasible in practice, we cast initialization as a *graph edge ranking* problem: assign an edge rank $r_{ij}$ to every pair and select a sparse, well-connected subset as $\mathcal{E}_0 = \text{Select}(r, \text{budget})$ that is likely to lead to an accurate 3D reconstruction.

Subsequent steps (outside the scope of this work) perform local matching and geometric verification *only* for edges in $\mathcal{E}_0$ (e.g., essential/fundamental/homography estimation with RANSAC) to recover

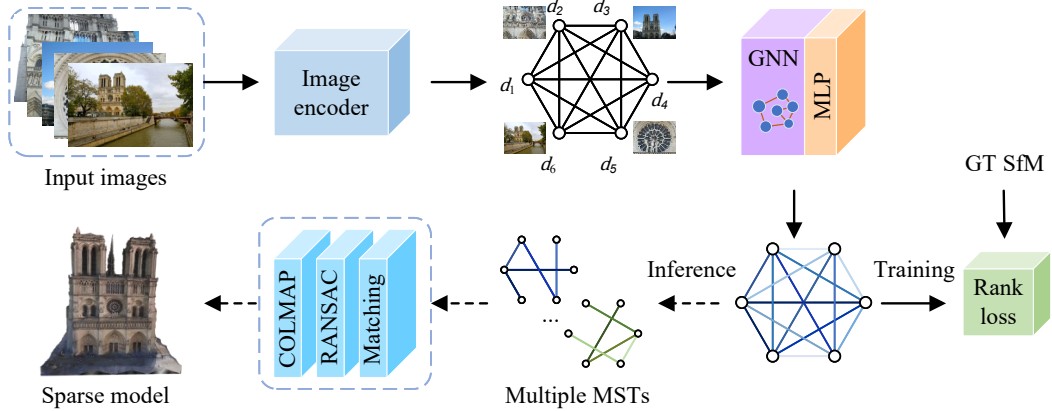

Figure 2: Overall pipeline, including a fine-tuned image encoder (DINOv2 backbone plus SALAD aggregator, etc), complete graph construction, GNN-based global matchability prediction for training with a ranking-based loss function supervised by 3D information. At inference, the edges are ranked by the predicted matchability and used to build multiple minimal spanning trees and the initial pose graph. Incremental SfM is applied on the pose graph and returns the final sparse 3D model.

relative poses $\{(R_{ij}, t_{ij})\}$ and produce the verified pose graph $\mathcal{G}_{\text{pose}} = (\mathcal{V}, \mathcal{E})$ with $\mathcal{E} \subseteq \mathcal{E}_0$ that is then further processed by the employed SfM method. Our contributions target the edge ranking and selection that determine $\mathcal{G}_0$; the rest of the pipeline remains unchanged.

## 3.2 EDGE RANKING PREDICTION

**Image Encoder.** Each image $I_i$ is encoded into a descriptor $d_i = f_{\text{en}}(I_i) \in \mathbb{R}^d$. A naive edge rank could be computed via cosine similarity $\langle d_i, d_j \rangle$ (this is what traditional methods essentially do), but such pairwise scores ignore the global structure of the dataset. We therefore augment embeddings with global reasoning using a GNN.

**Graph Neural Network.** Given the image embeddings $\{d_1, d_2, \ldots, d_N\}$, we follow the approach of (Turkoglu et al., 2021) and construct a complete graph $\mathcal{G}$ with the embeddings $d_i$ as $N$ nodes. Overall, the graph contains $N(N-1)/2$ directed edges $e_{ij}$, each initialized by the concatenation of the end node embeddings and an edge attribute derived from their pairwise similarity. Specifically, edge features are computed via a fully connected layer $f_l$ with ReLU activation on the end nodes:

$$e_{ij} = \text{ReLU}(f_l[d_i, d_j, \text{attr}_{ij}]), \tag{1}$$

where $\text{attr}_{ij} = \langle d_i, d_j \rangle$ and $e_{ij} \in \mathbb{R}^{256}$. We apply two iterations of a 2-GNN message passing module, with shared parameters across all edges. In the $t$-th iteration, each edge is updated using the initialized edge feature and the features of its end nodes as follows:

$$e_{ij}^t = f_{\text{edge}}([e_{ij}, d_i^t, d_j^t]), \tag{2}$$

where $f_{\text{edge}}$ consists of two linear layers with batch normalization and ReLU activations. Next, for each node $i$, messages from all neighbors $j$ are computed using:

$$m_{ji}^t = f_{\text{msg}}([e_{ij}^t, d_j]), \tag{3}$$

where $f_{\text{msg}}$ shares the same architecture as $f_{\text{edge}}$ but uses different input dimensions. The messages are then aggregated by averaging over all neighbors as follows:

$$m_i^t = \frac{1}{N} \sum_{j=1}^{N} m_{ji}^t. \tag{4}$$

Node features are updated by another two-layer MLP with ReLU activation, as follows:

$$d_i^{t+1} = f_{\text{update}}([d_i^t, m_i^t]), \tag{5}$$

The updated node and edge features from iteration $t$ are used as input for the next iteration. We perform two iterations. Finally, the edge features from the second iteration are passed through a two-layer MLP with ReLU activation and dropout (dropout = 0.5) to predict global edge rank:

$$r_{ij} = f_{\text{MLP}}(e_{ij}^{t=2}). \tag{6}$$

**Geometry-based Supervision.** To bridge the gap between generic representation learning and our target task, we supervise the model with signals derived from SfM. Given a set of training images, we run keypoint detection, matching, and relative pose estimation via RANSAC to connect the images locally and initialize a pose graph. For each image pair $(i, j)$, the number of RANSAC inliers $u_{ij}$ serves as the first source of supervision:

$$u_{ij} = \#\{\text{RANSAC inliers for pair } (i, j)\}. \tag{7}$$

While RANSAC is robust, the estimated relative poses may still be noisy or completely incorrect. The pose graph is therefore filtered, discarding pairs with $u_{ij}$ below a threshold, before being fed into SfM. As a complementary signal, we also consider the final 3D reconstruction. Let $v_{ij}$ denote the number of common triangulated 3D points observed by both images $i$ and $j$ after SfM:

$$v_{ij} = \#\{\text{3D points visible in both } I_i \text{ and } I_j\}. \tag{8}$$

This provides a second measure of geometric consistency. Both signals are normalized to $[0, 1]$ and combined by range normalization and averaging as follows:

$$\tilde{r}_{ij} = \tfrac{1}{2}\left(\text{norm}(u_{ij}) + \text{norm}(v_{ij})\right), \tag{9}$$

where $\tilde{r}_{ij}$ denotes the ground-truth edge score used for ranking in the supervision. The range normalization is done by mapping the values below 1000 to [0.0, 0.8] while the others to [0.8, 1].

Importantly, this scheme is *self-supervised*: $u_{ij}$ and $v_{ij}$ are only needed during training, and can be extracted automatically from any SfM pipeline without manual labels. At inference, our model predicts edge ranks directly from image embeddings and global context, without $u_{ij}$ and $v_{ij}$.

**Loss Function.** In contrast to works that train image encoders with categorical or binary labels (Berton & Masone, 2025; Izquierdo & Civera, 2024), our supervision is continuous. We care about the *relative* ordering of image pairs, not the exact value of their labels. Thus, we train the model as a ranking rather than a regression problem.

The quality of a predicted ranking is commonly evaluated in information retrieval with Normalized Discounted Cumulative Gain (NDCG) (Järvelin & Kekäläinen, 2002). DCG measures the quality of a ranked list by summing item relevance, discounted for lower-ranked items. For ground-truth ranks $r_i \in \{1, \ldots, M\}$ ($r_i = 1$ for top-1), the relevance score of item $i$ is $v_i = M - r_i$. Given predicted edge ranks, pairs are sorted, and for each item $i$ with predicted rank $\hat{r}_i$, DCG is computed as

$$\text{DCG} = \sum_{i=1}^{M} \frac{2^{v_i} - 1}{\log_2(\hat{r}_i + 1)}. \tag{10}$$

The ideal DCG (IDCG) is computed from the ground-truth ranking, and NDCG is defined as NDCG = DCG/IDCG, ranging from 0 to 1.

Since NDCG is non-differentiable, we adopt NDCGLoss2++ (Wang et al., 2018), built upon the LambdaRank algorithm (Burges et al., 2006; Burges, 2010), which optimizes a differentiable approximation of NDCG by considering pairwise item swaps and their effect on ranking quality. Its detailed formulation is provided in (Wang et al., 2018). This loss has been shown effective for training MLP-based ranking models (Pobrotyn et al., 2020). The only hyperparameter is $k$, which determines the number of items sampled per list, set to half the list size in our training.

### 3.3 EDGE SELECTION DURING INFERENCE TIME

**Selection with Minimum Spanning Tree (MST).** The predicted edge ranks $r_{ij}$ guide the selection of image pairs, *i.e.* graph edges. A common practice in the literature is to use $k$-Nearest Neighbors ($k$-NN) search for each image (Schonberger & Frahm, 2016; Pan et al., 2024), selecting the top-$k$ most similar images per query. However, $k$-NN selection considers only local neighborhoods and ignores global connectivity. To obtain a sparse yet informative set of image pairs for accurate Structure-from-Motion (SfM), we instead adopt Minimum Spanning Trees (MSTs). An MST is defined as the set of edges that connects all vertices with the minimum total weight. For $N$ images, an MST contains $N - 1$ edges, where each edge weight is defined as follows:

$$w_{ij} = 1 - r_{ij}, \tag{11}$$

with $r_{ij}$ denoting the predicted edge rank for pair $(i, j)$. MST-based selection outperforms $k$-NN in our experiments, as it guarantees global connectivity while prioritizing high-confidence edges. We compute MSTs using Kruskal's algorithm (Kruskal, 1956).

**Multiple MSTs.** While MSTs have been widely used for pose graph initialization – ensuring global connectivity with minimal cost (Schonberger & Frahm, 2016; Pan et al., 2024) – the use of *multiple MSTs* to provide redundant yet diverse connections is, to our knowledge, largely underexplored. Recent works such as Xiao et al. (2021) and Gan et al. (2024) employ unions of multiple MSTs or orthogonal trees, but these approaches rely on fixed similarity heuristics rather than learned signals.

In contrast, our approach integrates learned edge ranks into a multi-MST construction. Given the complete graph with predicted ranks $\{r_{ij}\}$, we iteratively build $k$ MSTs as follows:

(1) Compute the first tree as $\mathcal{T}_1 = \text{MST}(\{w_{ij} = 1 - r_{ij}\})$.

(2) For $m > 1$, penalize edges already selected in $\mathcal{T}_1, \ldots, \mathcal{T}_{m-1}$ by assigning infinite cost, then compute $\mathcal{T}_m$.

(3) Form the initial pose graph as the union of all trees, $\mathcal{G}_{\text{init}} = \bigcup_{m=1}^{k} \mathcal{T}_m$.

This construction guarantees at least $k$ disjoint paths between any two nodes, providing redundancy that improves robustness, since some edges may later be discarded by geometric verification. The first tree is optimal for the original graph, while subsequent trees are optimal given previously selected edges, thereby promoting diverse connectivity. Unlike $k$-NN selection, which may oversample dense local neighborhoods, the multiple-MST strategy enforces both global connectivity and structural diversity. To our knowledge, this is the first formulation that combines multiple MSTs with learned edge ranking for SfM initialization.

**Graph Clustering.** To scale to large image collections, we employ a graph clustering strategy (Chen et al., 2020; Damblon et al., 2025). The complete image graph is partitioned into smaller subgraphs before being processed by our GNN model for edge rank prediction. This partitioning prevents GPU memory exhaustion when handling large graphs. Following Damblon et al. (2025), we use METIS (Karypis & Kumar, 1997) for graph partitioning, guided by similarities derived from intermediate features of our encoder before the GNN. Predictions from different subgraphs are then aggregated to recover the final edge ranking matrix for the entire dataset.

## 4 EXPERIMENTS

We evaluate our approach on large-scale SfM benchmarks, focusing on the quality of the initial pose graphs and their impact on final reconstructions. Experiments compare multiple MSTs guided by different image embeddings against our learned edge rankings.

**Datasets.** Training is performed on the training set of MegaDepth (Li & Snavely, 2018), where we construct geometry-based ground-truth edge ranks as described in equation 9. For evaluation, we test on held-out scenes from MegaDepth as well as on 15 Phototourism scenes from the Image Matching Challenge 2023 (IMC23) (IMC, 2023). To assess robustness in highly ambiguous environments, we additionally evaluate on VisymScenes (Xiangli et al., 2025), a dataset of doppelganger images containing repeated patterns and strong structural symmetries.

**Backbone and Features.** We use the recent MegaLoc (Berton & Masone, 2025) as the backbone encoder, which combines frozen DINOv2 features with a SALAD aggregator and linear projections. On top of this encoder, we train our GNN-based edge ranking predictor. For tentative correspondences in COLMAP, we use SuperPoint (DeTone et al., 2018) with LightGlue (Lindenberger et al., 2023) matcher on MegaDepth and IMC23. On VisymScenes, following the practice in Xiangli et al. (2025), we adopt SIFT (Lowe, 2004) with brute-force matching (Bradski, 2000).

**Baselines.** We compare with state-of-the-art global retrieval methods, including CNN-based Cos-Place (Berton et al., 2022), AnyLoc (Keetha et al., 2023), DINOv2-based DINOv2-SALAD (Izquierdo & Civera, 2024), and MegaLoc (Berton & Masone, 2025). All methods are evaluated by feeding them into the default COLMAP pipeline. Importantly, we do not use the camera intrinsics provided for the datasets and let COLMAP estimate them, following the most practical scenario.

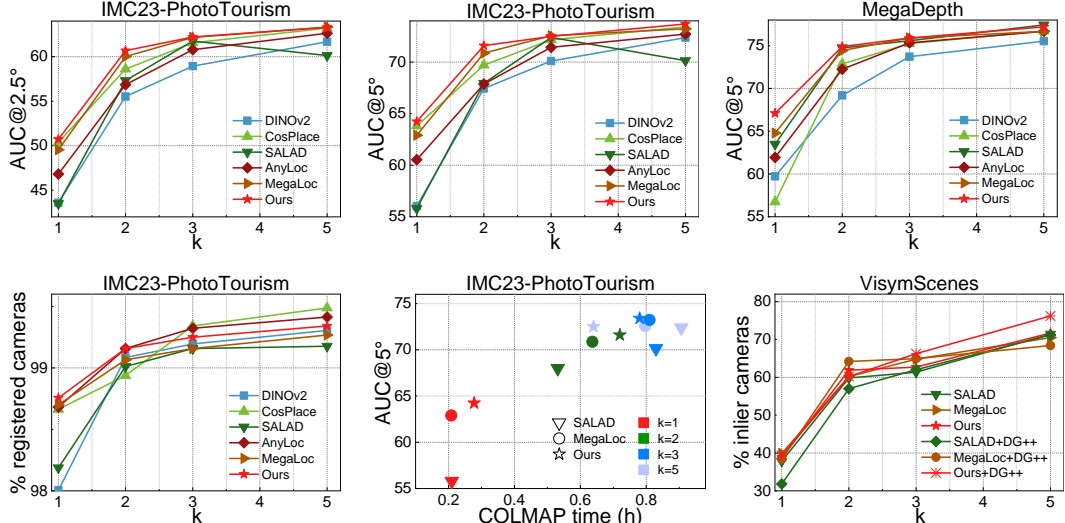

Figure 3: COLMAP reconstruction performance on pose graphs built from multiple MSTs guided by learned embedding scores or our edge ranks. On the first row, we show AUC@2.5° (*left*), AUC@5° (*middle*) on IMC23-PhotoTourism and Megadepth (*right*) datasets with an increasing number (*horizontal* axis) of MSTs. On the second row, the percentage of registered cameras (*left*) and COLMAP time vs. AUC@5° (*middle*) on IMC23 are shown. The last plot shows the ratio of accurately reconstructed cameras on the challenging VisymScenes dataset containing doppelgangers.

**Training Details.** The model is trained on 153 training scenes from MegaDepth, with 8 validation scenes for checkpoint selection. Each batch consists of 240 images sampled from a single scene, with a maximum of 4 batches per scene. The ranking loss is computed on the top half of edges according to ground-truth ranks. Training is performed for 50 epochs with AdamW (Loshchilov & Hutter, 2019), using a learning rate of $1 \times 10^{-5}$. We train on NVIDIA Tesla A100 GPUs (40GB, CUDA 12.4), while COLMAP is executed on CPU-only machines.

**Evaluation Protocol.** Following common practice in SfM, we evaluate reconstructions by the accuracy of relative camera poses between *all* estimated cameras. Specifically, we report Area Under the Recall Curve (AUC) at $2.5°$ and $5°$, the percentage of successfully registered cameras, and COLMAP runtime, as in Pan et al. (2024). Results are reported for different numbers of MSTs ($k \in \{1, 2, 3, 5\}$) used for graph construction.

On VisymScenes, we reconstruct four test scenes using $\{1, 2, 3, 5\}$ MSTs for COLMAP after also applying Doppelganger++ (DG++) filtering (Xiangli et al., 2025). To ensure stability, we fix the number of RANSAC iterations to 10k and use a fixed random seed. Unlike DG++, which normalizes inlier counts only among registered cameras, we normalize by the total number of geo-tagged images, penalizing unregistered views and providing a fairer evaluation.

**Scalability via Graph Clustering.** For large collections ($N > 500$), we partition the full image graph into subgraphs using METIS (Karypis & Kumar, 1997), following Damblon et al. (2025). The number of clusters is set as $n_{\text{clusters}} = 1 + \lfloor N/N_{\text{max}} \rfloor$. To preserve cross-cluster consistency, each subgraph is expanded to include direct neighbors of its nodes, excluding duplicates. Predictions from overlapping pairs are averaged, yielding the final global edge ranking matrix.

## 4.1 3D RECONSTRUCTION

We show experimental results on COLMAP reconstructed cameras on IMC23-PhotoTourism, MegaDepth, and a newly proposed, challenging duplicate dataset, VisymScenes.

**IMC23-PhotoTourism.** In Fig. 3, the first two columns of the row report the relative pose AUCs (at multiple thresholds) for reconstructions initialized with $k$ minimal spanning trees ($k \in \{1, 2, 3, 5\}$) on the IMC23 dataset. The test set contains 15 PhotoTourism scenes with an average of 360 images per scene (ranging from 68 to 911). Our method achieves the highest accuracy already at $k = 1$, providing strong initialization and further improving as more trees are added. CosPlace is the

| Method | Filter | AUC@5° ↑ | | | Time (min) ↓ |
|---|---|---|---|---|---|
| $k$ MSTs | → | 1 | 2 | 3 | 5 | 5 |
| SALAD | | 37.8 | 59.8 | 61.4 | 71.4 | **4.96** |
| MegaLoc | - | **39.9** | 60.2 | **64.8** | 70.5 | 7.29 |
| Ours | | 38.8 | **61.9** | 62.7 | 71.7 | 5.68 |
| SALAD | | 31.8 | 57.0 | 62.0 | 71.2 | **5.78** |
| MegaLoc | DG++ | 38.4 | **64.2** | 64.9 | 68.4 | 8.62 |
| Ours | | **39.3** | 60.2 | **66.3** | 76.2 | 7.43 |

Table 1: Percentage of inlier cameras on Visym-Scenes. Best results (per group) are in bold; ultimate best underlined. COLMAP mapping time at $k = 5$, in minutes, is shown in the last column.

| Variant | AUC@5° ↑ | | | |
|---|---|---|---|---|
| $k$ MSTs → | 1 | 2 | 3 | 5 |
| Ours w/ SALAD backbone | 61.2 | 68.3 | 72.4 | 70.7 |
| Ours w/o GNN | 55.4 | 70.2 | 72.0 | 72.5 |
| Ours | 64.2 | 71.6 | 72.5 | 73.7 |
| Oracle-RANSAC inliers | 65.7 | 72.5 | 73.0 | 74.1 |
| Oracle-3D inliers | 65.4 | 72.1 | 73.4 | 74.3 |
| Oracle* | 66.7 | 72.2 | 73.6 | 73.9 |

Table 2: AUC@5° scores on PhotoTourism using different components in the proposed approach shown. The last three rows show the comparison of two sources of supervision.

strongest competitor at $k = 1$, but its performance plateaus when additional trees are included. Pretrained MegaLoc ranks consistently second-best overall, while SALAD-based initialization fails on one scene at $k = 5$, with its best results achieved at $k = 3$.

The left plot in the second row of Fig. 3 show the percentage of registered cameras. For small $k$, our method registers more cameras than competing approaches, resulting in a stronger base graph for subsequent MSTs. At $k = 5$, CosPlace and AnyLoc register the largest number of cameras, but their poses are less accurate as discussed earlier.

The middle plot reports AUC@5° versus COLMAP runtime, with colors denoting different $k$ values and markers indicating competitors. Our method is slightly slower at $k = 1$ and $k = 2$, but achieves higher accuracy with only a minor runtime overhead. At $k = 3$ and $k = 5$, the trend changes: our approach becomes the fastest while also achieving the best accuracy. We attribute this to differences in graph quality: for small $k$, stronger graphs register more cameras, increasing runtime; for larger $k$, all methods register similar numbers of cameras, but our higher-quality edges make reconstruction more efficient. Overall, these results demonstrate that our global edge ranking produces more accurate and reliable pose graphs, particularly in the sparse regime.

**MegaDepth.** The top-right plot of Fig. 3 shows AUC@5° on MegaDepth test scenes "0015" and "0022" as $k$ increases. On average, each scene contains ∼500 candidate images. Our method consistently outperforms all baselines for small $k$, achieving the highest accuracy at $k = 1, 2, 3$. Only at $k = 5$ does SALAD slightly surpass our performance, but its accuracy drops considerably at $k = 1$, highlighting instability. Other baselines gradually improve with larger $k$, converging to similar AUCs, but remain well below our method in the sparse regime. These results confirm that our approach is particularly effective when the pose graph must remain highly sparse.

## 4.2 Disambiguating SfM

To evaluate the robustness of our method under severe visual ambiguities, we test on VisymScenes, a dataset containing challenging *doppelganger* images with highly similar but non-overlapping content. The benchmark provides test image pairs for four scenes, each containing roughly equal proportions of positive pairs and negative doppelgangers, resulting in image pools averaging 352 cameras.

For pose graph initialization, we select edges via MSTs weighted by either SALAD or MegaLoc similarity scores, or by our global edge predictions over the complete graph. After geometric verification, we further apply Doppelganger++ (DG++) (Xiangli et al., 2025) to remove spurious pairs. Thus, the final pose graph is constructed from MST-selected, RANSAC-verified edges, with optional DG++ filtering. As VisymScenes contains many doppelganger images, COLMAP consistently produces multiple disconnected components. We therefore evaluate each component separately by counting cameras reconstructed within a fixed threshold of their ground-truth geolocations.

Results are in Tab. 1 and in the bottom right plot of Fig. 3. Our method achieves the highest number of correctly localized cameras at $k = 2$ and $k = 5$, and these gains are further amplified when combined with DG++. Our method with DG++ achieves the highest accuracy by a significant margin of 4.6 percentage points. This highlights that the proposed global approach successfully complements the local decisions made by the DG++ filter. Importantly, our predictor is applied *before* RANSAC, reducing wasted computation on unpromising pairs. We show the COLMAP time at the highest scores of $k = 5$ in the last column. Our method is slower than SALAD, yet achieves much higher AUCs both with and without DG++ filtering. Finally, we note that VisymScenes spans diverse cities and

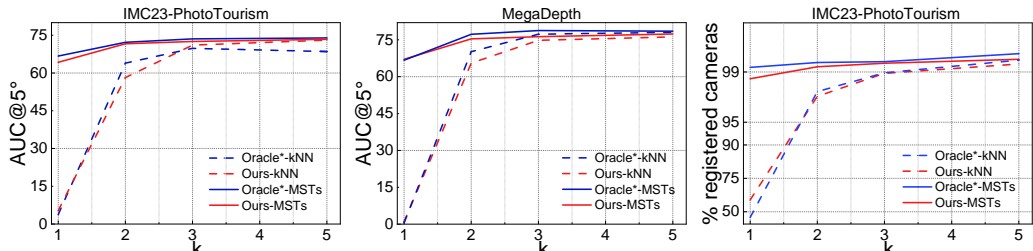

Figure 4: Relative pose AUC@5° of COLMAP on the IMC2023, PhotoTourism (*left*) and MegaDepth (*middle*) datasets using either $k$NN or $k$ minimal spanning trees as proposed in this paper for pose graph initialization. The number of registered cameras is also reported, in percentages (*right*).

scenarios beyond the landmark-centric settings common in SfM benchmarks. Despite this distribution shift, our model performs strongly without retraining, underscoring its generalization ability.

### 4.3 ABLATIONS

**Edge Selection.** We compare our multi-MST-based edge selection against the standard $k$-nearest neighbors ($k$NN) strategy, evaluating relative pose accuracy across all camera pairs. In Fig. 4 (left two plots), we report results on PhotoTourism (*left*) and MegaDepth (*middle*) test scenes. Using only the first nearest neighbor per image leads to poor connectivity, and consequently much lower accuracy, compared to the first minimal spanning tree. MST ensures that all images are connected into a single component. Extending this idea, combining multiple MSTs provides complementary edges, achieving higher accuracy while requiring fewer candidate pairs for geometric verification and subsequent optimization. The ratio of registered cameras is reported on the *right*, showing much more cameras are registered through MST than direct neighboring each image.

For reference, we also report oracle performance on both datasets, where edges are selected according to the number of overlapping triangulated 3D points or RANSAC-verified inliers. These results highlight that MST-based selection is not only more efficient than $k$NN but also aligns more closely with oracle preferences, leading to more accurate and compact pose graphs.

**Training.** Table 2 presents ablations of different components of our method. We first replace the MegaLoc backbone with SALAD. As expected, using the weaker backbone decreases accuracy slightly, although the results remain stronger than using SALAD off-the-shelf, which achieves $55.8$ AUC@5° (Fig. 3). This highlights that the proposed method is agnostic to the employed backbone. Removing the GNN leads to a large performance drop in the sparsest regime ($k = 1$), while the gap narrows when more edges are included ($k > 1$). This confirms the importance of the proposed GNN module for reasoning under limited connectivity.

Finally, we ablate the supervision signals. The last three rows in Tab. 2 report oracle rankings based solely on (i) RANSAC relative pose inliers and (ii) common 3D inliers after reconstruction. RANSAC inliers emphasize local pairwise connectivity and yield accurate reconstructions at small $k$, while common 3D inliers are more effective at larger $k$. The combined supervision (Oracle*), used throughout our experiments, provides the most consistent performance across all regimes.

### 5 CONCLUSION

We introduce the concept of edge prioritization for robust pose graph initialization in Structure from Motion. Our method addresses the limitations of traditional retrieval-based approaches by predicting globally consistent edge ranks using a visual encoder backbone and message passing in GNN with geometry-derived supervision. This strategy effectively bridges multiple images in the collection, leading to more reliable SfM. Building on the predicted ranks, we propose a multi-spanning-tree-based selection strategy that produces sparse yet well-connected graphs, showing superior performance on publicly available SfM datasets as well as duplicates. Especially on the extremely sparse case where limited edges are used, our approach consistently improves the reconstruction while being efficient. These results show that integrating global reasoning at the graph construction stage opens the door for faster and more reliable SfM pipelines. The source code and trained models will be made public.

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

## A  APPENDIX

Large Language Models (LLMs) are used for polishing the text.

