# OpenReview forum: "Global-Aware Edge Prioritization for Pose Graph Construction in SfM"
_ICLR.cc/2026/Conference — ICLR 2026 Conference Withdrawn Submission_

### Official Review · Reviewer_3Lad · 2025-10-27

**Soundness:** 2
**Presentation:** 3
**Contribution:** 2
**Rating:** 4
**Confidence:** 4

**Summary:**

The paper proposes a GNN-based approach for constructing pose graphs in SfM, replacing the traditional k-NN image retrieval step. The GNN ranks image pairs by their global importance for reconstruction, using geometry-derived supervision. The final pose graph is built from multiple minimum spanning trees. Experiments on MegaDepth, IMC23, and VisymScenes show improved reconstruction accuracy and robustness compared to existing retrieval-based baselines.

**Strengths:**

The paper is clearly written and well-structured, making the technical ideas easy to follow. Introducing a GNN for global reasoning in pose graph construction is an original and conceptually appealing idea. Overall, the work presents an interesting direction with a solid methodological contribution, though the empirical validation is limited.

**Weaknesses:**

There are several aspects that limit the strength of the contribution:

1. The claimed benefit of incorporating global context, particularly for handling duplicates as illustrated in Fig. 1, is not convincingly supported by the quantitative results. The method still relies on external Doppelganger filtering, which suggests that global reasoning alone does not effectively resolve such ambiguities.
2. The experimental evaluation is relatively limited in scope and size, considering that the main technical novelty is a new heuristic. Including more diverse and large-scale datasets such as MegaScenes (ECCV 2024) would provide stronger evidence of generalization.
3. Important ablation is missing, the number of GNN message-passing iterations, as well as an analysis of computational complexity and runtime scalability are missing.
4. The reported performance improvements over baselines like MegaLoc are relatively modest, hence the practical significance of the proposed approach is somewhat limited.

**Questions:**

- Can the authors provide quantitative evidence that the proposed global reasoning helps disambiguate duplicates, beyond the qualitative examples in Fig. 1?

- What is the computational overhead of the GNN-based ranking compared to standard k-NN retrieval, and how does it scale with large image sets?

- Would it make sense to measure the quality of the constructed graphs directly?

---

### Official Review · Reviewer_XCTu · 2025-10-28

**Soundness:** 3
**Presentation:** 2
**Contribution:** 2
**Rating:** 2
**Confidence:** 4

**Summary:**

This paper aims at enhancing the robustness of SfM methods via ranking edges in view graphs. It proposes to use a GNN to learn globally connected information, where edges are ranked via the trained GNN. During inference, the edge ranks are used to construct k-MSTs, which can be useful in discarding potentially wrong edges. The GNN is trained on the MegaDepth datasets. The method is evaluated on the Visym-Scenes and PhotoTourism datasets.

**Strengths:**

(1) The task and idea to rank edges in view graph is interesting and novel.
(2) The method provide a simple yet (seems) promising method to discard potentially wrong matches in view graph, thus can enhance the robustness of SfM methods.
(3) The paper provides quantitative results on the Visym-Scenes dataset and the PhotoToursim dataset to verify the effectiveness of the method.

**Weaknesses:**

(1) The implementation details of the model is unclear. e.g. the feature dimension for the node features, edge features and global feature, which makes the reimplementation of the method difficult.
(2) Important baselines are missing in the paper (COLMAP, GLOMAP, etc.).
(3) While the paper claims it can used for enhancing the robustness of SfM under ambiguous scenes, no qualitative results are provided in the paper.
(4) Qualitative results are missing in the paper. Moreover, though the camera pose accuracy provides quantitative metrics for reference, ground-truth camera poses for the evaluated datasets could be inaccurate. Another choice is to provide novel view synthesis quality as complementary metrics (as is done in ACE0 [ECCV 2024]).

**Questions:**

1. At line 320, the authors said they compare their method to SOTA global retrieval baselines. But it is confusing to me how these baselines are used in the experiments (e.g. Do they simply used for constructing view graphs?) It must be made clear in the paper and explained in the rebuttal.

2. I would like to know the generalizability of the GNN for ranking edges in view graphs. To me, the training dataset is not large enough, and the validated scenes are quite limited. More results on more diverse scenes should be provided (e.g. 7scenes, tanks-and-temples, etc).

---

### Official Review · Reviewer_XC7n · 2025-10-29

**Soundness:** 2
**Presentation:** 3
**Contribution:** 3
**Rating:** 6
**Confidence:** 3

**Summary:**

This paper introduces a global-aware edge prioritization framework to improve pose graph initialization in Structure-from-Motion (SfM). Traditional methods rely on local, pairwise image retrieval, which often misses critical connections. The authors propose a Graph Neural Network (GNN) to predict edge ranks by leveraging global context from the entire image set, supervised by 3D geometric data. This is combined with a multi-minimum spanning tree (MST) selection strategy to build sparse yet robust pose graphs. Experiments on benchmarks like MegaDepth and IMC23 show significant gains in reconstruction accuracy, especially under sparse graph conditions.

**Strengths:**

- Clear problem reframing & conceptual novelty. Recasting pose-graph initialization as global edge ranking (not independent retrieval) is a crisp, consequential shift. The GNN aggregates set-level context; multi-MST selection enforces global connectivity while promoting structural diversity.
- Grounded supervision & principled objective. The supervision signal combines RANSAC inliers with common 3D points from an SfM run—self-supervised, label-free, and task-aligned—optimized via a differentiable NDCG surrogate (LambdaLoss/NDCGLoss2++). This is a thoughtful fit to ranking.
- Strong results in the sparse regime. The method notably improves AUC and registered cameras when k is small (e.g., k=1–3), precisely where sparse graphs are most fragile and practical speed demands are highest.
- Challenging disambiguation study. On VisymScenes, the approach complements DG++, increasing correctly localized cameras and demonstrating robustness to doppelgangers, a real failure mode for SfM.

**Weaknesses:**

- Scope of baselines at the graph construction stage. While retrieval backbones (CosPlace, AnyLoc, DINOv2-SALAD, MegaLoc) are compared as sources of scores, it remains unclear how the proposed global ranker stacks up against pairwise overlap predictors / re-rankers explicitly designed to suppress non-matchable pairs before verification (e.g., overlap-based voting or pairwise matchability models). The related-work section cites such directions (e.g., VOP/overlap prediction), but end-to-end comparisons at graph-building time are limited.
- Dependence on backbones. The best results use MegaLoc (DINOv2 + SALAD) features. Although the method is backbone-agnostic in principle, the practical gains may attenuate under lighter encoders; the ablation with SALAD suggests some degradation. A more systematic latency/accuracy trade-off across encoders would be valuable.
- High-speed / Sparse Regime. The paper highlights improvements under budget-limited, very sparse pose-graph initialization, instantiated by varying the number of MSTs k. Using k as a proxy conflates sparsity with compute/time budgets and lacks a standardized, method-agnostic definition; as a result, cross-method fairness and external validity remain unclear.
- Policy & disclosure nit. Appendix notes that “LLMs were used for polishing the text,” which is fine as disclosure, but ICLR typically expects clarity on what content (if any) was machine-generated and whether any evaluation artifacts were affected (they were not, per the paper). A brief compliance note would remove any ambiguity.
- Citation standards. To further improve readability, please consider the citation formatting adjustments. (e.g. line 051: “but only pruned Wilson & Snavely”; line 192: “the approach of (Turkoglu et al., 2021)”)

**Questions:**

- k vs. budget guidance. In practice, how should users set the number of MSTs k versus a total edge budget? Could you provide a simple heuristic mapping from desired geometric-verification budget to k (or a stopping rule based on rank margin / connectivity)?
- Rank calibration & stability. The weight uses 𝑤𝑖𝑗=1−𝑟𝑖𝑗. How sensitive is MST construction to monotone transforms of 𝑟𝑖𝑗? Have you considered pairwise uncertainty (e.g., predictive variance) to down-weight brittle edges during tree construction?
- Broader baselines at the construction stage. Could you include overlap-prediction / matchability baselines that re-score candidate pairs prior to RANSAC, run under the same MST framework (i.e., replace 𝑟𝑖𝑗)? This would disentangle benefits from (a) global reasoning and (b) MST selection itself.
- Runtime composition. The time-accuracy plots are helpful; could you break runtime into (rank prediction + MST building + COLMAP) to quantify how much of the gain comes from fewer wasted verifications vs. cleaner graphs?

---

### Official Review · Reviewer_2DZW · 2025-10-30

**Soundness:** 2
**Presentation:** 2
**Contribution:** 2
**Rating:** 2
**Confidence:** 5

**Summary:**

Large-scale SFM retrieves images based on image similarity, computes epipolar geometry among the retrieved images for downstream reconstruction tasks. This paper argues that this traditional method overlook the global cues at the time of retrieval and can lead to loss of information for correct initialisation.
Hence this paper proposes a GNN based edge ranking method on the SfM derived supervision to predict the edge ranks. These edge ranks are used for further downstream objective of SFM to select the correct pairs required for reconstructions. Such pairs are geometrically more relevant for the reconstruction task.

**Strengths:**

The paper provides a key insight on the initialisation problem of SfM and gives a fresh thought on what lacks in the initial view graph construction.
The paper attempts to solve the said problem and shows the improved matching results in complicated pairs of images which consist of large view point changes.
The paper isn easy to follow.

**Weaknesses:**

There are number of weaknesses in this paper.
The major weakness is the method is not tried on creating large-scale SfM results which was the initial problem from where the sub problem is derived. Hence the utility of this method in the context of SfM is not clear.
In line 53”limiting the quality of later refinement” is a statement which is not evaluated by this method. Generally global SfM produces better reconstruction in divide and conquer based SfM methods and the refinement is possible if any images are missed. The paper does not show what is the significance of this problem in the context of SfM and how much this method is contributing to that.
If we use SfM supervision then how we can generalise to other dataset? RANSAC also a random process.

With no results on SfM, the other weaknesses and the paper current condition leads to significant need of improvement before publishing the paper.

**Questions:**

1. It is not clear how fl is trained in equation 1.
2. Does e_{ij} is fixed over iteration?
3. Eqn 7,8,9 will produce different values for different runs. How to select which run output to take?
4. Line 244-245 is not clear

---

### Official Review · Reviewer_xzzm · 2025-11-01

**Soundness:** 2
**Presentation:** 2
**Contribution:** 1
**Rating:** 2
**Confidence:** 4

**Summary:**

This paper presents the method of Edge Prioritisation or ranking to initialise the Pose Graphs for the global structure-from-motion pipeline, moving beyond per-image ranking used traditionally. The key idea is to assign a rank to edges or image pairs based on its utility or expected contribution to the SfM process. They use a Graph Neural Network (GNN) to assign the ranks, which is trained on SfM-derived dataset. Thാബ also uses multiple spanning trees in the initialisation process.

**Strengths:**

Creating a global rank to prioritise edges (between pairs of images) seems to be a useful, new idea. This eliminates the use of similarity-based selection of images to initialise the pose graph. A representation learned from the training set of MegaDepth is employed using GNNs.

**Weaknesses:**

- The method is not motivated or explained well. What is the intuition behind defining the GNN as they are defined. I don't get the basis of defining the graph structure with $f_l$, $f_{edge}$, $f_{update}$, $f_{MLP}$, etc.? Why those and why not something else?
- Results do not appear to be significant compared to prior work.

**Questions:**

I have a number of questions/doubts/comments/observations on the work as presented. Each may not require a response from the authors, but represent the basis of my decision/apprehensions.
- While GNNs are good tools in this setting, can this method be called self-supervised (L235)? Yes, $u_{ij}$ & $v_{ij}$ are available using prior SfM methods on the standard dataset, but then those become the effective "ground truth", which will define an upper-bound on performance, won't they?
- Equations 1, 2, 3, 5, 6 define the components of the Graph Neural Networks used in this work. I certainly want to know the motivation and intuition behind these designs. Neural networks are black-boxes by themselves; the best we can do in practice is to explain the choices made and establish them using appropriate evaluations of the chosen parameters through ablations. This work falls short on that count and appears ad-hoc overall.
- Cosine similarity is all that is used by traditional methods (L188). In L197, $f_l$ uses the individual embeddings and their cosine as input. How does this add to the traditional method?
- How did the dimnesonality of 256 come for $e_{ij}$? What is the dimensionality of the embeddings $d_i$?
- Does $f_{MLP}$ directly predict the edge rank $r_{ij}$ as given in Eq 6, with lower numbers representing better matches?
- For $r_{ij}$ given in Eq 9, it would seem higher is better (more inliers is good) which will be like an edge-score and not an edge--rank. Is there some confusion here, which may be just notational? Particularly in the light of the para in L240-242.
- The loss used is not clear to me. What is $\hat{r}_i$ in Eq 10? Is it computed by sorting r values from Eq 9?
- IDCG iis computed from ground truth ranking, using  $r_i$ in place of  $\hat{r}_i$, if I understand correctly. However, $v_i$ and $r_i$ are related. That would make IDCG a constant, won't it? Is that the intention?
- Eq 11 will suggest higher the better for $r_{ij}$ as MST uses minimum of $w_{ij}$. Is that correct?
- Use of Multiple MSTs: If I get it right, Xiao (2021) and Gan (2024) also used them. The novelty of this work is in using them with learned signals. Why is this so significant (L293)?
- Is it true that the multi-MST approach guarantees k disjoint paths? Later MSTs may involve the infinite weight edges given non-dense connectivity, won't they? What is the impact of that?
- The use of graph clustering is mentioned (L295). What is its impact? Does the procedure need to construe the $n^2$ graph first before partitioning? Do they need all $e_{ij}$ values?
- Fig 3: Unfortunately, I don't see a clear win for the proposed method over the SOTA. Performance seems adequate at k=3 also. Results here are mixed at best. Most methods are very similar at k=3. Why is k=5 required at all? All seem to plateau there.
- Regarding COLMAP time: Why is there a big reduction in the proposed method for k=5 from k=3? Explanation around L400 isn't satisfactory.
- Table 1 results are also mixed, big advantage is only with k=5 and DG++, but not in run time!
- Fig 4: 1-NN as a baseline for comparison is not acceptable as nobody will use it.
- I didn't understand the relevance of the Oracle scores in Fig 4 and Tab 2.

---

### Note · Authors · 2025-11-14

I have read and agree with the venue's withdrawal policy on behalf of myself and my co-authors.